# Novel Perspectives on the Design and Development of a Long-Acting Subcutaneous Raltegravir Injection for Treatment of HIV—In Vitro and In Vivo Evaluation

**DOI:** 10.3390/pharmaceutics15051530

**Published:** 2023-05-18

**Authors:** Heba S. Abd-Ellah, Ramesh Mudududdla, Glen P. Carter, Jonathan B. Baell

**Affiliations:** 1Medicinal Chemistry, Monash Institute of Pharmaceutical Sciences, Parkville, VIC 3052, Australia; hebasaleh.chem@gmail.com; 2Medicinal Chemistry Department, Faculty of Pharmacy, Minia University, Minia 61519, Egypt; 3Microbiology and Immunology Department, Peter Doherty Institute for Infection and Immunity, The University of Melbourne, Parkville, VIC 3001, Australia; glen.carter@unimelb.edu.au; 4School of Pharmaceutical Sciences, Nanjing Tech University, No. 30 South Puzhu Road, Nanjing 211816, China

**Keywords:** HIV, long-acting subcutaneous injection, raltegravir, cyclisation-activated prodrugs

## Abstract

Antiretrovirals (ARVs) are a highly effective therapy for treatment and prevention of HIV infection, when administered as prescribed. However, adherence to lifelong ARV regimens poses a considerable challenge and places HIV patients at risk. Long-acting ARV injections may improve patient adherence as well as maintaining long-term continuous drug exposure, resulting in improved pharmacodynamics. In the present work, we explored the aminoalkoxycarbonyloxymethyl (amino-AOCOM) ether prodrug concept as a potential approach to long-acting ARV injections. As a proof of concept, we synthesised model compounds containing the 4-carboxy-2-methyl Tokyo Green (CTG) fluorophore and assessed their stability under pH and temperature conditions that mimic those found in the subcutaneous (SC) tissue. Among them, probe **21** displayed very slow fluorophore release under SC-like conditions (98% of the fluorophore released over 15 d). Compound **25**, a prodrug of the ARV agent raltegravir (RAL), was subsequently prepared and evaluated using the same conditions. This compound showed an excellent in vitro release profile, with a half-life (t_½_) of 19.3 d and 82% of RAL released over 45 d. In mice, **25** extended the half-life of unmodified RAL by 4.2-fold (t_½_ = 3.18 h), providing initial proof of concept of the ability of amino-AOCOM prodrugs to extend drug lifetimes in vivo. Although this effect was not as pronounced as seen in vitro—presumably due to enzymatic degradation and rapid clearance of the prodrug in vivo—the present results nevertheless pave the way for development of more metabolically stable prodrugs, to facilitate long-acting delivery of ARVs.

## 1. Introduction

HIV remains a global health problem. It affects 36.7 million people worldwide, with an additional 2.1 million new cases diagnosed each year, indicating that the AIDS pandemic is far from over [1]. Nevertheless, the introduction of antiretroviral (ARV) treatment and prophylaxis measures has led to remarkable progress in defeating HIV/AIDS infection. These agents have significantly improved treatment efficacy, reduced HIV-related morbidity and mortality, and increased the lifespan of HIV-positive individuals to a level comparable to that of the HIV-negative population [2,3,4]. However, lifelong compliance with the daily dosing required by ARV regimens remains a great challenge, especially in HIV patients with coexisting mental conditions or drug abuse, or for whom there are multiple medications to be taken daily, complicating the treatment regimen [5,6]. Even short-term non-adherence to ARVs can result in therapeutic failure, as well as promoting development of drug-resistant HIV viral strains that decrease the patient’s future treatment opportunities [7]. In fact, only 62% of HIV patients maintain the ≥90% adherence required for optimal viral suppression [8]. Furthermore, despite the high efficacy of pre-exposure prophylaxis (PrEP) in preventing HIV transmission, PrEP compliance ranged from 22–98% in clinical trials [9], also indicating low adherence for a significant number of individuals. In such circumstances, long-acting ARV injections requiring weekly or monthly administration might be an optimal, life-saving alternative to daily oral therapy.

In a 2013 survey, more than 80% of 400 adult patients indicated they would probably try long-acting ARV therapy if the dosage was once a month [10]. In view of this, a number of long-acting injections are currently in phase 2 and phase 3 clinical trials for HIV treatment and prevention. For instance, a phase 2b clinical trial demonstrated that long-acting intramuscular injection of two ARVs (rilpivirine and cabotegravir), administered every 4 or 8 weeks, was as effective as daily oral therapy in maintaining viral suppression over 96 weeks [11]. Two subsequent phase 3 clinical trials on the same drug combination reached equivalent conclusions [12,13]. In 2021, the FDA approved Cabenuva as the first intramuscular long-acting formulation to treat HIV. It is supplied as a packaged kit of two separate long-acting injectable medicines, cabotegravir and rilpivirine, which should be administered once every two months [14,15].

Long-acting injections such as these utilise various approaches to provide sustained therapeutic drug exposure over a period of weeks, or sometimes longer. These include, but are not limited to: incorporation of the active ingredient in either an oil-based solution or a suitable matrix from which the drug is slowly released; using a nanosuspension of drug particles; microsphere encapsulation; implants; and in situ forming depots [16,17,18,19,20,21,22,23]. Each of these approaches has its unique strengths and weaknesses; for example, persistent pain at the injection site that can last for more than three months, low drug loading capacity with subsequently high administration volume, conformational changes, and others. Although many efforts have been directed towards improving the currently available approaches, no reports have yet addressed the potential of chemical transformation methods as a means to develop long-acting injections. Herein, we present our initial work on the amino-alkoxycarbonyloxymethyl (amino-AOCOM) ether moiety as a pH-activated handle to develop long-acting prodrugs for HIV treatment.

Previous work demonstrated that amino-AOCOM prodrugs **1** and **2** (Figure 1) release the parent drug via a pH-dependent intramolecular cyclisation reaction that is completely independent of enzymatic biotransformation [24]. Both prodrugs exhibited enhanced aqueous solubility and oral bioavailability compared with the parent compound [24]. Recently, we applied the same strategy to develop mesalamine prodrugs **3** and **4**—with three- and four-methylene spacers, respectively (Figure 1)—for colonic delivery [25]. In that study, a key finding was that increasing the methylene spacer length to five or six carbons led to pH-sensitive linkers with very slow-release profiles. These were of limited interest for colonic delivery, but could be useful in long-acting injectable formulations. In this paper, we describe our initial efforts to evaluate the utility of those extended handles in such a context. The first part of our study focused on the use of 4-carboxy-2-methyl Tokyo Green (CTG) fluorophore as a model compound, to establish the initial proof of concept. In part two, we applied the same approach to raltegravir (RAL), as a representative ARV drug utilised in the treatment of AIDS. Mechanistically, RAL inhibits the HIV integrase enzyme to block integration of viral DNA into the human genome [26]. The half-life (t_½_) of RAL is 7–12 h, allowing for a regimen of two 400 mg doses per day [27]. It also exhibits highly desirable pharmacokinetic properties, such as rapid distribution to the female genital tract and cervicovaginal fluid, as well as greater penetration into the seminal compartment [28,29,30]. In 2016, a long-acting RAL formulation was developed, involving reconstitution of milled, γ-irradiated drug in a sterile vehicle of water containing poly(ethylene glycol) 3350 (5%), polysorbate 80 (0.2%), and mannitol (5%). Preclinical studies showed that two weeks after a single subcutaneous (SC) injection in BLT mice (7.5 mg) and rhesus macaques (160 mg), the plasma concentration of RAL was comparable to that achieved through the usual oral regimen in humans [28]. Accordingly, it is considered a promising long-acting ARV and prophylactic agent, thereby rationalising its choice in the current study.

## 2. Materials and Methods

### 2.1. Synthetic Procedures and Analytical Data

All chemical reagents and solvents were obtained from Combi-Blocks (San Diego, CA, USA), Sigma Aldrich (Burghausen, Germany), and Thermo Fisher Scientific (Waltham, MA, USA), kept in appropriate storage conditions, and used without further purification. Reactions were monitored by thin layer chromatography (TLC) using silica gel 60 F254 coated aluminium plates with 0.25 mm thickness. TLC plates were visualised under UV light at 254 nm or 366 nm, and staining with KMnO_4_ solution when necessary. The ^1^H NMR and ^13^C NMR spectra were obtained at 400.13 MHz, 100.62 MHz, respectively, on Bruker spectrometer (Bruker Corporation, Billerica, MA, USA), using tetramethylsilane as an internal reference. Chemical shifts were measured in parts per million (ppm) and referenced to an internal standard of residual deuterated solvent: CDCl_3_ (7.26 ppm for ^1^H and 77.16 ppm for ^13^C), DMSO-d_6_ (2.50 ppm for ^1^H and 39.52 ppm for ^13^C), or MeOD (3.31 ppm for ^1^H and 49.00 ppm for ^13^C). Missing or/and overlapping ^13^C signals were identified using 2D NMR experiments (HSQC and HMBC) and are distinguished with a * symbol. The ESI–MS and HRMS analyses were run on Agilent UHPLC/MS (1260/6120) and Agilent 6224 TOF-MS systems (Agilent Technologies, Santa Clara, CA, USA), respectively. The purity of the final target compounds was determined by analytical HPLC (Agilent 1260 Infinity, Agilent Technologies, Santa Clara, CA, USA) and was >95% in every case.

### 2.2. In Vitro Release Tests

Probes **14**–**15** and **20**–**21** stock solutions were prepared at 1000 µM (1 mM) in a 1:1 mixture of MeCN/water, followed by serial dilution with deionised water to give a 50 µM working concentration. Each experiment was run in triplicate after dilution of 0.5 mL of the stock solution with 0.5 mL of PBS (pH 7.4) or Tris buffer (pH 7.4), giving a final assay concentration of 25 µM for the probes. The samples were incubated at 34 °C to mimic the conditions found in SC tissue. Every day, a 2 µL aliquot of each sample was subjected to HPLC analysis using an Agilent 1260 Infinity instrument, with conditions as follows. Column: Agilent ZORBAX Eclipse Plus C18 Rapid Resolution, 3.5 μm (4.6 × 100 mm), pore size (95 Å); column temperature (35 °C). The mobile phases were solvent A: 0.1% TFA in ultrapure H_2_O and solvent B: 0.1% TFA in MeCN. The injection volume was 2 μL and the flow rate was kept at 1.0 mL/min. The analysis was performed using a gradient elution of 5–100% of solvent B in solvent A over 9 min, then maintained for a further 1 min at two wavelengths: 254 nm and 214 nm. The retention times of CTG and probes **14–15** and **20–21** were 4.13, 4.27, 4.93, 5.55, and 5.71 min, respectively. A calibration curve for the free CTG fluorophore was constructed over the concentration range of 10–100 µM at pH 7.4. The data were fit by linear regression to the equation y = 14.615x − 0.067, where y is the peak area of the analyte and x is the analyte concentration (µM). The correlation coefficient (R2) was computed as 0.999, confirming the linearity of used method under the specified measured concentrations.

Likewise, a test solution of prodrug **25** was incubated in 50% PBS at 100 µM final concentration, and analysed using the same HPLC method but with detection at 254 nm only. A higher concentration of **25** was used due to its relatively weaker absorption at 254 nm, compared with CTG. The retention times of RAL and prodrug **25** were 4.47 and 5.55 min, respectively. A calibration curve of RAL was constructed over the range 50–1000 µM. The data were fit by linear regression as before to give the equation y = 5.1961x − 20.064, with a similar goodness of fit (R^2^ ≥ 0.999).

### 2.3. Pharmacokinetic Study of Prodrug **25**

#### 2.3.1. Study Design

The pharmacokinetic profile of prodrug **25** was evaluated in mice, in a study undertaken at WuXi AppTec. Prodrug **25** was dissolved in a sterile vehicle of DMSO/Solutol HS15/water (10:10:80) to give a clear solution with a final concentration of 6 mg/mL. The same procedure was used to prepare a 6 mg/mL solution of RAL as a control. Six male BALB/c mice were used, each weighing 18–25 g and aged between 7–9 weeks (three per group). A single 30 mg/kg dose of prodrug **25** was administered to each animal in the treatment group (subcutaneously, in the thoracic region). Similarly, RAL (30 mg/kg) was administered to each animal in the control group. Blood samples were taken at the time points indicated below, up to the experiment endpoint (96 h).

#### 2.3.2. Blood and Sample Preparation

For each animal, blood samples (30 μL each) were collected from the submandibular or saphenous vein over a period of 96 h, at the following time points: zero/pre-dose; then 0.25, 0.5, 1, 2, 3, 4, 5, 6, 24, 48, and 96 h. All samples were transferred into pre-chilled commercial K2EDTA tubes and placed on wet ice until processing for plasma, which was obtained by centrifugation at 4 °C and 3200× *g* for 10 min, within 30 min of sampling. The plasma samples were transferred to polypropylene tubes, snap-frozen on dry ice, and kept at −70 ± 10 °C until LC-MS/MS analysis.

#### 2.3.3. Data Analysis

An LC-MS/MS method was developed to measure the concentrations of prodrug **25** and RAL in the biological matrix. Two mobile phases were used: mobile phase A (0.025% NH_4_OH and 2 mM NH_4_OAc in 95:5 *v/v* water/MeCN) and mobile phase B (0.025% NH_4_OH and 2 mM NH_4_OAc in 5:95 *v/v* water/MeCN). A UPLC method was performed using an LC-MS/MS-BM Triple Quad 6500 plus system with an ACQUITY UPLC BEH C18 1.7 μm column (2.1 × 50 mm) at a flow rate of 0.65 mL/min. The assay time was 2.5 min, and the solvent B composition was as follows: 10–100% (0.2–1.4 min), 30% (1.41–1.7 min), 95% (2.0–2.4 min), and 10% (2.41–2.5 min). The retention times for the prodrug, RAL and the internal standards (tolbutamide and verapamil) were 0.76, 1.04, 0.66, and 0.97 min, respectively. Quantification was performed by monitoring the transition of *m*/*z* 835.3 [M+H]^+^ to 457.2 for prodrug **25**, representing the cyclisation of the linker with the amidic RAL such as in side-product **22**; whereas for RAL, the transition *m*/*z* 445.2 [M+H]^+^ to 361.2 was followed, representing the loss of the 2-methyl-1,3,4-oxadiazole moiety as the main fragmentation pathway. For the internal standards, the *m*/*z* transition from 455.2 [M+H]^+^ to 164.9 was used for verapamil (corresponding to formation of the verapamil fragment [CH_2_CH_2_-(3,4-dimethoxy phenyl)]^+^), and for tolbutamide, the transition of *m*/*z* of 271.1 [M+H]^+^ to 155.1 was used, representing formation of the [CH_3_-Ph-SO_2_]^+^ fragment. The LLOQ was found to be 1 ng/mL for **25** and 0.3 ng/mL for RAL, respectively. The method was linear over the measured standard concentrations for **25** and RAL. The mean plasma concentration was determined for **25** and RAL (as metabolite or control) at each sampling time point. Pharmacokinetic parameters were then calculated by non-compartmental analysis of plasma concentration/time profiles using the Phoenix WinNonlin 6.3 software. The computed parameters were as follows: peak plasma concentration (C_max_); the time over which the initial concentration is reduced to half (t_½_); area under the plasma concentration versus time curve from time zero to 96 h (AUC 0–96 h); and mean residence time from time zero to 96 h (MRT 0–96 h).

#### 2.3.4. Ethics

Animals were obtained from an approved vendor (Beijing Vital River Laboratory Animal Technology Co., Ltd., Beijing, China), and were acclimated for at least 3 d before being placed in the study. Animals were housed with free access to water and food at all times. The study was approved by the WuXi Institutional Animal Care and Use Committee on Animal Experiments, and was performed in an AAALAC-accredited laboratory, complying with PHS policy and national regulations on the administration of laboratory animals.

## 3. Results and Discussion

### 3.1. Proposed Release Mechanism

The proposed mechanism of drug release from the designed prodrugs, based on fundamental principles and supported by the related literature, is shown in Figure 2. Here, the prodrugs become less protonated at physiological pH (7.4), rendering the amino group relatively more nucleophilic and causing subsequent intramolecular cyclisation–elimination to release the payload along with side-products **5** (cyclic carbamate) and **6** (formaldehyde), neither of which is toxic at low concentration [24,31]. The length of the methylene spacer dictates the size of the ring formed, and hence, the rate of cyclisation. Prodrug **1** reportedly cyclises rapidly to give a five-membered ring, releasing >90% of the parent compound within 1 h at pH 7.0. In contrast, prodrug **2** cyclises more slowly to form a six-membered ring, releasing >55% of the payload over 5 h at the same pH [24]. Increasing the methylene spacer length to four carbons—such as in prodrug **4**, which we reported recently [25]—hinders the cyclisation further and slows down drug release accordingly (in this case, achieving 92% release of mesalamine over 6 d at pH 6.5).

Therefore, in the case of RAL (where long-acting prodrugs are required), we tried extending the methylene spacer to five or six carbons (corresponding to eight- and nine-membered ring cyclisations) in our initial model probes, to obtain the very slow-release kinetics suitable for development of a long-acting SC injection. This goal also necessitates an understanding of the physiology of the SC tissue, also known as the hypodermis, and which is mostly composed of connective tissue, separated primarily by adipose tissues, and to a minor extent by macrophages and fibroblasts. The fibroblasts produce constituents of the extracellular matrix (ECM), including glycosaminoglycans (GAGs), elastin, and collagen [32,33]. In general, a drug after injection reaches the interstitial space of the hypodermis, where absorption occurs mainly according to its molecular size. For instance, smaller molecules (≤16 kDa, as defined in this cited study), are preferentially absorbed through the vascular endothelium of blood capillaries due to their unrestricted permeability, whereas macromolecules (>16 kDa) and small particles are absorbed through the peripheral lymph vessels from the surrounding interstitial space, before entering the systemic circulation [34,35]. However, many other factors can influence the absorption process, rendering it complex and unpredictable. These factors are either physiological (e.g., interaction of the drug with endogenous compounds, lymph flow, or blood), or physicochemical (e.g., hydrophilicity and electrostatic charge) [36]. In the current approach, the prodrug should ideally be retained in the SC tissue, with systemic absorption of only the released payload. In an attempt to decrease the permeability of the prodrug across the vascular endothelium, a phosphonic acid group was introduced via click chemistry using an alkyne group incorporated into the AOCOM structure. Phosphonic acids ionise at physiological pH, resulting here in a highly charged prodrug with presumably poor permeability [37,38]. In support of this hypothesis, phosphonic acid-containing drugs reportedly show impaired diffusion across biological membranes and require endocytosis for cell penetration [37,38,39]. Nevertheless, given the lack of a reliable in vitro model to predict SC bioavailability, it was difficult to predict whether the phosphonic acid group would play this role in vivo, and we were interested to ascertain whether or not it is useful in such a context.

In addition, due to its anionic nature at physiological pH, this functionality is also reported to increase the water solubility of polymers [38], organic compounds, and ligands for co-ordination chemistry [40]. A phosphonate should therefore improve the aqueous solubility of the current prodrugs, which is a clear requirement for an SC formulation, to avoid any precipitation in vivo that might occlude capillary vessels [41].

### 3.2. Synthesis

To synthesise probes **14**–**15** and **20**–**21**, the AOCOM iodides (**11a**–**b**) were first constructed as illustrated in Figure 1. In brief, Boc-protected amino alcohols (**9a**–**b**) were reacted with chloromethyl chloroformate to give AOCOM chlorides **10a**–**b**, which were subsequently converted into the corresponding iodides **11a**–**b** via a Finkelstein reaction as illustrated in Appendix B. *tert*-Butyl-CTG **12** was obtained as described previously [42,43], which was then reacted with **11a**–**b** in the presence of Cs_2_CO_3_ to give the desired CTG conjugates **13a–b**. These intermediates (**13a–b**) were deprotected under acidic conditions (aq. HCl) to afford final probes (**14**–**15**) in good yields (Figure 1).

To introduce the phosphonic acid moiety, bromophosphonate **16** was used as starting material, and was reacted with NaN_3_ in DMSO to give azido ester **17** in high yield (Figure 2). We noted that in a previous study, the phosphonic acid azide was preferred over the equivalent phosphonate diester for the click reaction, as subsequent dealkylation of the latter (after triazole formation) failed to give the desired product [44]. Therefore, compound **17** was first dealkylated using bromotrimethylsilane followed by methanolysis, to provide azide **18**. Direct coupling of CTG probes **14**–**15** with **18** was attempted via Cu (I)-catalysed azide/alkyne cycloaddition at room temperature, but the reaction failed and unreacted starting material remained. Further, this was tried at 70 °C using a microwave reactor, however all the starting materials profoundly decomposed, and the free fluorophore was generated. To resolve this issue, an alternative Boc-protected compounds **13a**–**b** were used to access probes **20**–**21**, since they are relatively more stable than probes **14**–**15** at elevated temperatures. Thus, **13a**–**b** were reacted with azide **18** to give intermediates **19a**–**b**, which were deprotected under acidic conditions to give the final probes **20**–**21**, as shown in Figure 2.

Similarly, RAL was alkylated with AOCOM iodide **11b** using Cs_2_CO_3_ (3 equiv.) and DMF as solvent. An unexpected side-product with an *m*/*z* of 457.1 [M+H]^+^ was formed during the reaction, and was identified as **22** based on NMR analysis as shown in Figure 3, and Appendix A. It was assumed that alkylated product **23** was formed initially but underwent subsequent intramolecular cyclisation via the amidic NH of RAL to give **22**. To avoid this side-reaction, the coupling conditions were optimised with respect to limiting equivalents of Cs_2_CO_3_ and reaction time. Using a stoichiometric amount of Cs_2_CO_3_ (1 equiv.) and 1 h reaction time resulted in clean reaction conversion to the desired product **23**, without any traces of side-product **22**. Increasing the number of equivalents of Cs_2_CO_3_, and/or extending the reaction time promote the base induced cyclisation and side-product formation.

Next, **23** was reacted with azide **18** to give compound **24**, which was again deprotected under acidic conditions to afford prodrug **25**. Interestingly, the major product formed during this hydrolysis step was identified as **26**, based on its NMR spectra and observed *m*/*z* of 853.3 [M+H]^+^, as demonstrated in Appendix B and shown in Appendix A. This product was believed to arise through hydrolysis of the oxadiazole moiety of **25** in aqueous acidic medium, as shown in Figure 3. A similar tendency for the RAL oxadiazole to hydrolyse has been reported, when it was stirred in a mixture of aqueous KOH and MeCN [45]. Using 2 M HCl in diethyl ether/MeCN as an alternative to aqueous acid, prodrug **25** was obtained in good yield.

### 3.3. In Vitro Release Study of Probes **14–15** and **20–21**

In vitro Release assays for probes **14–15** and **20–21** were conducted using pH and temperature conditions similar to those of SC tissue (pH 7.4 and 34 °C) [46]. Test compounds were dissolved in either MeCN/PBS (1:1, pH 7.4) or MeCN/Tris buffer (1:1, pH 7.4) at 25 µM, then incubated at 34 °C. Aliquots were taken daily and analysed by HPLC and LC-MS until complete fluorophore release was achieved. HPLC was preferred over fluorometric-based assay to study the stability of the synthesised probes because of the bleaching effect that may occur to the fluorophore over the long analysing periods. Photobleaching is the inability of the fluorophore to fluoresce due to the fluorophore instability upon repeated exposure to light over long periods [47]. It can be a particular problem for fluorescence-based assays, thereby rationalising the choice of HPLC in the current work.

The results showed that probe **14**, with a five-methylene spacer, released 36% and 58% of the CTG fluorophore after 1 d and 2 d, respectively, in 50% PBS. Further payload release was slow, reaching 98% after 9 d (Figure 4A). In comparison, probe **15**—with a six-methylene spacer—appeared to be more stable than **14** at every time point. Under the same conditions, it released 29% and 50% of CTG after 1 d and 2 d, respectively, and this value reached 98% after 10 d. Release of CTG from probes **20**–**21** was then measured to study the influence of the phosphonic acid group on the kinetic profiles. These experiments indicated that CTG release was slower with the phosphonic acid group attached, in both cases. For example, probe **20** gave 25%, 41%, and 98% release of CTG after 1, 2, and 13 d, respectively (Figure 4A). Payload release became much slower with the longer spacer in probe **21**, with 20%, 33%, and 98% of CTG released after 1, 2, and 15 d, respectively, as shown in Figure 4A.

Evaluation of the release kinetics of all CTG probes indicates that they exhibited first-order kinetics. The release rate constants (*k*_obs_) and half-lives (t_½_) were determined and are shown in Table 1. The results showed that *k*_obs_ for probe **14**, with a five-methylene spacer, was 0.41 d^−1^, while it was 0.35 d^−1^ for probe **15**, with a six-methylene spacer; thereby confirming our earlier report that the length of the methylene spacer is a key parameter in the release process. Furthermore, attaching the phosphonic acid group to the AOCOM handles further decreased the reaction rate for reasons that are currently not clear (*k*_obs_: 0.41 and 0.35 d^−1^ for **14**–**15** versus 0.24 and 0.22 d^−1^ for **20**–**21**, respectively). The half-lives were 1.69, 1.98, 2.88, and 3.15 d for probes **14**–**15**, and **20**–**21**, respectively, as summarised in Table 1.

Unexpectedly, the release of CTG from all probes was relatively faster in 50% Tris buffer at the same pH. Furthermore, probes **14**–**15** and **20**–**21** gave very similar CTG release rates: 89.2%, 86.2%, 82.5%, and 80.9%, respectively, after 1 d (Figure 4B), indicating a minimal effect of linker length on the release rate. It was assumed that the nucleophilic amino group of Tris had attacked the amino-AOCOM carbonate group to effect CTG release (Figure 4). This type of reactivity for Tris buffer was first reported in 2016, when an adduct formed by nucleophilic addition to asparagine succinimide-containing proteins was identified (Figure 4) [48].

Throughout this work, the identity of the fluorophore released from the probes (**14–15** and **20–21**) was confirmed by LC MS analysis in every case. Although the cyclised side product **5** was not separated to confirm the hypothesised mechanism, there are several reasons to believe that the proposed intramolecular cyclisation elimination reaction is the relevant mechanism. The literature reports the cyclisation of similar substrates [24,49,50,51,52], and the rate of fluorophore release decreased when the probes were incubated at lower pH (7.0, 5.0), supporting the hypothesis that the amino group plays a crucial role in the release mechanism. Moreover, cyclisation is reported to be slower with increasing methylene spacer length between the amino group and the carbonate group [24,25], and this also matched our results: probe **15**, with a six-methylene spacer, cyclised more slowly than probe **14**, with a five-methylene spacer. Based on the foregoing results, the pH-sensitive amino-AOCOM handle with a six-methylene spacer was chosen to develop the long-acting RAL prodrug (**25**).

### 3.4. In Vitro Release Study of Prodrug **25**

Similarly to the assays above, prodrug **25** (100 µM) was incubated in 50% PBS at 34 °C, and aliquots analysed daily by HPLC and LC-MS until full RAL release was achieved. The results indicated that 8.4% of RAL was released after 1 d, then this value increased slowly to reach 31.5% after 10 d, and finally plateaued at 82.1% after 45 d (Figure 5).

Although the identity of the released RAL was confirmed by LC-MS throughout the assay, it also became apparent that payload release was incomplete due to side-product formation as the experiment progressed. Analysis of LC-MS spectra indicated that the side-product formed was due to instability of RAL under the assay conditions, giving rise to **27** as shown in Figure 6. Side-product **27** was produced by hydrolysis of RAL’s 2-methyl-1,3,4-oxadiazole moiety, similarly to the formation of side-product **26**. The same result was obtained after incubation of unmodified RAL as a control under identical assay conditions. Oxadiazole hydrolysis can occur under acidic and neutral conditions, and reportedly also in basic media [45], confirming that the observed side-reaction can take place relatively easily and does not need strongly acidic or basic conditions to proceed. In this study, the degree of side-product formation was estimated as 15.2%, accounting for the unreleased RAL from **25**. Furthermore, maximal release of RAL from this prodrug was evidently prolonged compared with that for CTG from probe **21** (45 d versus 15 d).

The release profile of prodrug **25** at pH 7.4 indicates that it exhibited the same release pattern as the CTG probes; i.e., first-order kinetics. However, the reaction rate was significantly slower for probe **21** with the same methylene spacer (*k*_obs_ values were 0.036 d^−1^ for **25** versus 0.22 d^−1^ for **21**; Table 1). The t_½_ of **25** was prolonged accordingly, at 19.3 d compared with 3.15 d for **21**. CTG is more acidic than RAL (pK_a_ 4.33 versus 6.7) [53], and as illustrated in the literature for similar substrates, the rate of payload release increases with the payload’s acidity [52,54], thereby justifying the present findings.

### 3.5. Pharmacokinetic Release Profile of **25** in BALB/c Mice

Since in vitro models are not fully predictive of in vivo pharmacokinetics, especially for prodrugs, assessment of these characteristics still relies on in vivo studies [55]. Therefore, the pharmacokinetic profile of prodrug **25** was evaluated in mice. Compound **25** and the control (RAL) were dissolved in DMSO/Solutol HS15/water (10:10:80). For a single SC injection of **25** (30 mg/kg) in BALB/c mice (n = 3), the plasma concentrations of prodrug and released RAL were measured at the following time points: zero/pre-dose; then 0.25, 0.5, 1, 2, 3, 4, 5, 6, 24, 48, and 96 h. The plasma profile (Figure 7) was obtained by plotting the measured concentrations (ng/mL) against time (h). The same analysis was performed for the control group (n = 3), with administration of a single SC dose of RAL (30 mg/kg). The lower limit of quantification (LLOQ) was calculated as 1 ng/mL for **25** and 0.3 ng/mL for RAL. In both cases, the experiment continued until the concentration of the measured compounds had fallen below the LLOQ (96 h).

The results indicated rapid absorption of the RAL control, with the maximum plasma concentration (C_max_) of 6.23 µg/mL reached after 0.33 h (t_max_), as shown in Table 2. The concentration then declined quickly, with only 10.7 ng/mL remaining after 6 h (Figure 7B), and displaying an observed half-life of 0.76 h. These findings confirmed that RAL is rapidly absorbed and cleared in vivo following SC administration, illustrating the need to extend its half-life for parenteral formulations.

Unexpectedly, in case of doing with compound **25**, this prodrug was detected intact in plasma shortly after SC dosing, with the highest concentration (C_max_ = 4.55 µg/mL) found at the t_max_ of 0.25 h (Figure 7A), indicating that the prodrug was rapidly absorbed and was not retained in the SC tissue as designed. Furthermore, **25** was cleared quickly, with a t_½_ of 0.85 h. The released payload (RAL) was also measured in plasma, with C_max_ = 6.59 µg/mL and t_max_ = 0.42 h (Table 2). The released drug concentration then declined, with 1.3 ng/mL remaining at 48 h.

The mean exposure to RAL over the analysis period (AUC 0–96 h) was 5.59 µg h/mL for SC administration of prodrug **25**, compared with 9.89 µg h/mL for control administration of the unmodified drug by the same route (i.e., relative bioavailability = 56% of control exposure). This appeared to be related to the slightly extended release of RAL from prodrug **25**, since the t_½_ of the released drug was successfully increased (3.18 h versus 0.76 h for the control group; ~4.2 fold). Although these results were not as promising as the earlier in vitro tests (t_½_ = 19.3 d), they did provide initial proof of concept that the amino-AOCOM approach can successfully extend the half-life of RAL in vivo. At the same time, the significant difference between the in vitro and in vivo release profiles suggests that the prodrug may be susceptible to enzymatic hydrolysis, thus explaining why a steady plasma concentration of released RAL was not obtained.

In summary, the intention was that prodrug **25** should be retained in SC tissue, where it gradually breaks down through intramolecular cyclisation rather than enzymatic activation to give a slow but steady release of RAL over a long period. Unexpectedly, prodrug **25** was detected in plasma within a very short time-frame after dosing SC and cleared rapidly in vivo and cannot be practically used for the intended application. The significant difference between the in vitro and in vivo release profiles of **25** suggested that the major issue with the current design is the enzymatic instability of the prodrug in SC tissue or in plasma, and thus explaining why a steady plasma concentration of released RAL was not obtained. The other outstanding problem was the inability of the phosphonate to retain the prodrug in the SC tissue as designed. In terms of lack of retention in SC tissue, we thought it possible that electrostatic repulsion might have occurred between the negatively charged ECM and the negatively charged prodrug, leading to rapid absorption and transport of the prodrug via the blood capillaries. Although it may seem counterintuitive that a negatively charged species may be more rapidly absorbed, this assumption is supported by previous work in which negatively charged antibodies had enhanced bioavailability (70%) after SC injection relative to their positively charged counterparts (31%) [56,57], although this cannot be entirely confirmed due to the complexity of the absorption process from the hypodermis.

In order to improve the current design and to put the amino-AOCOM approach into practice, two challenges need to be overcome. Firstly, the instability of prodrug in vivo which could potentially be avoided by incorporating a bulky group into the carbon chain of the linker, as shown in Figure 8. This modification has been reported to lead to more enzymatically stable prodrugs [54]. The second challenge is to find a suitable way to achieve long-acting drug delivery in vivo as an alternative to the phosphonate group. In general, two approaches are mainly used to develop long acting-subcutaneous injection: prolonging SC drug retention (as aimed in the current work by using the phosphonate) and as reported in the literature through direct immobilization [58,59,60], and hydrogel formation [61,62], or extending the half-life of drugs via extending their blood circulation such as poly(ethylene glycol)ylation (PEGylation) [63]. We had many reasons to prefer PEGylation in future work to achieve our goal. Initially, PEGylation is the most common approach to delay in vivo clearance, by forming a hydration shell around the drug to increase its hydrodynamic radius and slow renal clearance [64]. One previous study demonstrated that PEGylation of an HIV fusion inhibitory peptide (C34) was able to decrease its clearance in vivo after SC administration and extend its half-life by 4.6-fold [65], in line with previous reports [66]. As we previously mentioned, prodrug **25** was able to extend the half-life of RAL by 4.2-fold, which is comparable to the PEGylation case just mentioned. Combining the amino-AOCOM-based approach with PEGylation would represent a novel strategy that could presumably lead synergistic extension of in vivo half-life of the prodrug, compared with either approach alone. However, the effect of PEGylation on the release rate of the drug (via the intramolecular cyclisation mechanism) would be an important parameter to address. Further, the proposed PEGylation approach should also confer protection against enzymatic degradation. It would also be preferable to retain the phosphonic acid functionality, since it improved the aqueous solubility of **25**. In brief, we believe that the high in vivo clearance of the prodrug and enzymatic instability can be decreased by PEGylation. The PEGylated, α-hydroxy, ω-phosphonic acid compound **28** features both phosphonic acid and hydroxy groups, to make it suitable for our intended future work. The hydroxy group can be converted into an azide functionality, as a handle for attachment to the prodrug via click chemistry (Figure 8).

The potential immunogenicity of the PEG moiety has recently been reported to increase the hydrophobicity of PEG end group [67]. Since compound **28** has a polar phosphonic acid end-group, we believe it should have lower immunogenicity, which we consider to be an extra advantage. Therefore, future efforts will be dedicated towards the synthesis of **29**, enabling us to study the effect of different bulky groups on the enzymatic stability of the prodrug (for example, using in vitro assays performed using isolated blood plasma). Furthermore, we intend to look for reliable in vitro models to study the effect of PEGylation on the prodrug’s clearance and correlate the results with subsequent in vivo findings.

## 4. Conclusions

A series of amino-AOCOM ether prodrugs have been designed and evaluated for long-acting SC delivery. Model compound **21**, with a six-methylene spacer, exhibited very slow-release kinetics in vitro (98% CTG released over 15 d) under pH and temperature conditions chosen to mimic those found in SC tissue. Based on this finding, RAL prodrug **25** was synthesised and evaluated in vitro and in vivo. Compound **25** degraded slowly in vitro to release RAL over 45 d (t_½_ = 19.3 d). On SC administration in mice, the plasma half-life of RAL was extended by 4.2-fold for **25** compared with the control drug (3.18 h versus 0.76 h), but this result was not as promising as the earlier in vitro observations. Thus, future optimisation is required to translate the early potential of this approach into long-acting in vivo delivery.

## Data Availability

All the data obtained within this work were provided in Appendix A.

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
