# Peer review of "Novel Perspectives on the Design and Development of a Long-Acting Subcutaneous Raltegravir Injection for Treatment of HIV—In Vitro and In Vivo Evaluation"

_pharmaceutics, 2023, doi:10.3390/pharmaceutics15051530_

Round 1

Reviewer 1 Report

Dear Author,

This is very interesting approach for prodrug design.

1.What was rational of using pro-drug approach instead for sustained release dosage form for translational prospects?

2.Which formulation barriers of  Raltegravir your hypothesis overcomes?

3.All NMR should be given code like methyl group A and should be presented on Spectra for easier understanding like you did for Figure S 42. 1 H NMR of 26 in MeOD?

4.Which functional group of Raltegravir essential for its activity ? Did you masked same group to block its prodrug character? 

5.You can change title of long acting as your release do not extent several weeks?

6.Test aqueous solubility of both parent and prodrugs because its BCS-2 class i expect solubility might improve?

3.

Scientifically English is ok.

Author Response

Dear Reviewer,

Thank you for taking your valuable time to review our research and providing your valuable suggestions to improve our manuscript. Please find the revised manuscript along with our responses to your comments below. We believe our responses and modifications to the original manuscript will meet your satisfaction and the original criteria for publication in Pharmaceutics.

Once again we would like to thank you from all the authors.

Sincerely, 

Corresponding authors, 

Reviewer 2 Report

The authors explored the amino-alkoxycarbonyloxymethyl (amino-AOCOM) ether prodrug concept as a potential approach to long-acting ARV injections. This work proposed some novel viewpoints. However, I still have some issues that the author needs to address, which are listed below:

1. In the pharmacokinetic study section , only 3 mice were used in each group, and the weekly age of mice were missing. Large sample size of experimental mice is expected.

2. It seems that there is an error for 3,200×g. Please revise it.

3. A more easy readable way for figure4b is expected, such as, column.

4. Does the side-product 27 of partial hydrolysis of RAL have cytotoxic? Does the side-product 27 affect the slow release of RAL?

5. In this paper, the author believes that the reason for drug precursor 25 rapidly enters the blood is its negative charge. Please further exclude other possibilities, such as the receptors on capillary endothelial cells for endocytic transport.

6. Please address the correlation between the length of carbon chain of the drug precursor 25, the cyclization rate and half-life.

7. The pharmacokinetic study is on the serum level of mice. Then what about the more general nephrotoxicity, hepatotoxicity and the toxicity of by-products.

Author Response

(The authors gave the same response as above.)

Reviewer 3 Report

Abd-Ellah et al, in their paper entitled “Design and Development of a Long-Acting Subcutaneous Raltegravir Injection for Treatment of HIV—in vitro and in vivo Evaluation”, present their data the development of a long-acting prodrug formulation of Raltegravir.  By adding an amino-alkoxycarbonyloxymethyl (amino-AOCOM) ether moiety as a pH-activated handle, the prodrug has improved solubility and oral bioavailability, and Raltegravir is released via a unique pH-dependent intramolecular cyclisation reaction.  Adding a phosphonic acid group in the moiety and manipulating methylene spacer length, led to pH-sensitive linkers with slow-release profiles, in vitro.   Unfortunately, in vivo the prodrug only had a modest 4-fold plasma half-life extension over RAL itself.  Paper is well written and nicely organized for the reader.  The authors are to be commended on their chemical pursuit for a long-acting RAL, and this data can be used to build upon for helpful drug design in the future. 

Prodrug 25 was best candidate in the study.

1.      Was the prodrug release ever monitored in T cell lines? How would that compare to it’s release in vitro?

2.      Was the IC50 ever determined in tissue culture cells?

3.      What is the CC50?

Minor:

1.      Section 3.3:  Title should be all italics to be consistent in paper

2.      Figure 8: capitalize “P” in proposed

3.      Figure S15 and S18:  unbold “probe” in title

4.      Figure S22 and S24:  remove word “probe” in title?

5.      Figure S25 and SS26: add the word ‘probe”?

6.      Figure S30:  bold “21” in title

7.      Figure S33: bold “22” in title

8.      Since Figure S41 has the word “prodrug” 25, add the word prodrug to S38-40

Author Response

Dear Reviewer,

Thank you for taking your valuable time to review our research and providing your valuable suggestions to improve our manuscript. Please find the revised manuscript along with our responses to your comments below. We believe our responses and modifications to the original manuscript will meet your satisfaction and the original criteria for publication in Pharmaceutics.

Once again we would like to thank you from all the authors.

Sincerely, 

Corresponding authors

Round 2

Reviewer 2 Report

Thanks for revision. No more comment.